# Oxidative Cyclization at *ortho*-Position of Phenol: Improved Total Synthesis of 3-(Phenethylamino)demethyl(oxy)aaptamine

**DOI:** 10.3390/md21050311

**Published:** 2023-05-19

**Authors:** Yuki Nakatani, Risa Kimura, Tomoyo Kimata, Naoyuki Kotoku

**Affiliations:** College of Pharmaceutical Sciences, Ritsumeikan University, 1-1-1 Noji-higashi, Kusatsu 525-8577, Shiga, Japan; ph0141xp@ed.ritsumei.ac.jp (Y.N.); ph0159ir@ed.ritsumei.ac.jp (R.K.); tomoyo0036kimata@gmail.com (T.K.)

**Keywords:** demethyl(oxy)aaptamine, total synthesis, oxidative cyclization, hypervalent iodine reagent, anti-mycobacterial agent

## Abstract

A shorter synthesis of the demethyl(oxy)aaptamine skeleton was developed via oxidative intramolecular cyclization of 1-(2-azidoethyl)-6-methoxyisoquinolin-7-ol followed by dehydrogenation with a hypervalent iodine reagent. This is the first example of oxidative cyclization at the *ortho*-position of phenol that does not involve spiro-cyclization, resulting in the improved total synthesis of 3-(phenethylamino)demethyl(oxy)aaptamine, a potent anti-dormant mycobacterial agent.

## 1. Introduction

Aaptamines are a class of alkaloids derived from marine sponges, particularly from *Aaptos* sp., with cytotoxic, antifungal, and antibacterial activities [1,2,3]. Recent studies disclosed the potential of aaptamine as therapeutic agent for neuropathic pain or Alzheimer’s disease [4,5], enhancing the importance of the compound. Demethyl(oxy)aaptamine (**1** in Figure 1) [6], an oxidized aaptamine derivative, is also known for its interesting biological activities. We have recently reported the potent antimicrobial activity of a related compound, 3-(phenethylamino)demethyl(oxy)aaptamine (PDOA, **2** in Figure 1) [7], against dormant *Mycobacterium bovis* BCG, with a minimum inhibitory concentration (MIC) value of 1.56 µM under both aerobic and hypoxic conditions. Compound **2** also exhibited potent anti-mycobacterial activities against drug-sensitive *M. tuberculosis* H37Rv and drug-resistant *M. tuberculosis* strains, with MIC values of 1.5–6.0 µM [8,9].

Recent anti-mycobacterial natural products studies focus on the search for compounds with novel mode of action (MOA) because of the emergence of drug resistant Mtb [10], and these results confirmed the potential of compound **2** as an anti-TB drug with a novel MOA. However, further studies such as in vivo assays and target identification are required. We have previously accomplished the total synthesis of **2** to overcome its supply limitations as a natural product [9]. In addition, we have reported a partial structure-activity relationship study of **2** through the synthesis of some analogs [11]. In this study, we present a more efficient synthetic route to obtain **2** and accelerate future biological studies.

## 2. Results and Discussion

### 2.1. Retrosynthesis

Since the first synthesis of aaptamine accomplished in 1985 [12], there are numerous reports on the total synthesis of aaptamines [3]. In most previous studies, including our previous report [9] and other recent excellent reports [13,14], the benzo[*de*][1,6]-naphthyridine core structure was first constructed from isoquinoline (AB) or quinoline (AC) rings (Figure 2A). Then, the two functional groups required to form the third ring were appended onto the two respective rings, introducing additional reaction steps and lowering the total yield. The alternative C-H oxidative cyclization of an aminoethyl side chain onto either ring circumvents these drawbacks to achieve a more efficient total synthesis of aaptamines.

The oxidative cyclization of phenol and phenol ether derivatives mediated by hypervalent iodine reagents is one of the most powerful and versatile methods for the synthesis of complex polycyclic compounds [15,16]. The aromatic ring is activated by the hypervalent iodine reagent, followed by intramolecular nucleophilic addition by the side-chain functional group at the *para*-position of the phenol or phenol ether. The scope of this reaction can be increased by performing the cyclization at the *ortho*-position of phenol (Figure 2B), although spiro-cyclization of *ortho*-substituted phenols has been well studied [17].

In this study, we improved the synthetic route of **2** by employing an oxidative cyclization step. Figure 1 shows our retrosynthetic approach. According to a recent report, the 2-phenethylamino side chain of **2** can be introduced into demethyl(oxy)aaptamine (**1**) at the final stage of the synthesis [14]. The precursor, **3,** can be used in an *ortho*-selective oxidative cyclization reaction to form the benzo[*de*][1,6]-naphthyridine ring. Azide groups are good nucleophiles in oxidative *para*-cyclization reactions for the synthesis of pyrroloiminoquinone alkaloids [18,19]. The introduction of a C2-unit and subsequent functionalization toward the C-1 position of the known isoquinoline **5** [20] is feasible.

### 2.2. Total Synthesis of 3-(Phenethylamino)demethyl(oxy)aaptamine

#### 2.2.1. Synthesis of Cyclization Precursors

First, the synthesis of compound **3**, the phenol-type demethyloxyaaptamine (**1**) precursor required for oxidative cyclization, is shown in Figure 2. Isoquinoline **5**, prepared using a previously reported method, was treated with *m*CPBA to yield the corresponding *N*-oxide **6**, which rearranged to afford isoquinolin-1-one **7** in moderate yield. Compound **7** was treated with trifluoromethanesulfonic anhydride in the presence of triethylamine to produce the *O*-triflate **8**. The side chain required for cyclization was introduced via Pd(dppf)Cl_2_-catalyzed Suzuki coupling between potassium vinyltrifluoroborate and compound **8** [21]. The use of microwave irradiation provided reproducible results with up to 78% yield. Subsequent introduction of an azide group at the side chain terminal of vinylisoquinoline **4** was accomplished using the modified azidation method reported by Zhou [22]. The original method was developed for the diazidation of alkenes, using 3.5 equiv. of TMSN_3_. In this work, we reduced the amount of TMSN_3_ to 1.5 equiv. to effectively form the hydroazidation product **9** in good yield. Finally, the precursor **3** was obtained in good yield via debenzylation with BCl_3_.

Second, the phenol ether-type precursor **15** was prepared (Figure 3). According to the similar method in Figure 2, commercially available 6,7-dimethoxyisoquinoline (**10**) was converted to the corresponding quinolin-1-one (**12**) [23]. Then, subsequent triflation and Suzuki-coupling with vinyltrifluoroborate provided compound **14** in moderate yield. Final hydroazidation afforded the desired precursor **15**, although relatively low reaction yield than Figure 2 was observed.

We also examined another method of 2-azidoethyl side chain elongation. Suzuki-Miyaura coupling reaction between triflate **13** and potassium β-(*N*-Boc)aminoethyltrifluoroborate (**16**), separately prepared through reported method [24], gave compound **17** in moderate yield. Subsequent removal of Boc group by TFA and diazotransfer reaction using imidazole-1-sulfonyl azide (**19**) [25] provided compound **15** in good yield. However, this procedure might not be superior to the above vinylation-hydroazidation method at the reproducibility and scalability.

#### 2.2.2. Oxidative Cyclization and Completion of Total Synthesis

Having obtained the desired precursors **3** and **15**, we examined their oxidative cyclization to demethyl(oxy)aaptamine (**1**). The detailed results of our optimization using phenol-type precursor **3** (Figure 4) are summarized in Table 1. The reaction of **3** with phenyliodine bis(trifluoroacetate) (PIFA) in CH_3_CN yielded trace amounts of a red compound with distinct UV absorption (Entry 1). Structural elucidation of the product revealed the expected cyclization at the *ortho*-position of the phenol and subsequent oxidative aromatization to **1** instead of the formation of the dihydro-analog **20**. Thus, subsequent experiments were performed using 2 equiv. of the oxidation reagents. The use of fluorinated alcohols as solvents improved the reaction yield, particularly when using 1,1,1,3,3,3-hexafluoro-2-propanol (HFIP) (Entries 2 and 3). Phenyliodine diacetate (PIDA) was more effective than PIFA as an oxidant and showed similar solvent effects (Entries 4–6). The reaction in higher concentration (0.1 M) provided various side products, probably through intermolecular coupling reaction, resulting in the lower yield, and the use of highly diluted reaction conditions (0.025 M) provided the best yields (up to 70%) (Entries 6–8). Other oxidants, such as iodoxybenzoic acid (IBX) and Cerium (IV) ammonium nitrate (CAN), were not effective (Entries 9–11).

Finally, **1** was treated with 2-phenethylamine in H_2_O under air, according to the reported condition [14], to afford PDOA (**2**) in a good yield. The spectral data of the product were identical to those reported in the literature.

A plausible mechanism for this oxidative cyclization reaction is shown in Figure 5. PIDA first reacts with the phenolic hydroxy group of **3**, followed by cycloaddition through nucleophilic attack of the azide group to the *ortho*-position of the phenol ring in intermediate **A**. Subsequent rearomatization and elimination of the diazo group affords intermediate **D**. Oxidative dehydrogenation of **D**, mediated by another PIDA molecule, yields the desired compound **1** through the formation of intermediate **E**. The treatment of phenol ether-type precursors, compound **9** or its methyl ether derivative **15** with PIDA did not yield any cyclization product analogous to **1** or **20**, resulting in complete recovery of the starting material. The addition of BF_3_·OEt_2_ or TMSOTf to activate PIDA was not effective. This suggests that the reaction does not proceed through single electron transfer (SET) from an aromatic ring of compound **3** [18,19], but through the direct phenol activation [26] and subsequent cyclization proposed here.

In summary, we developed an improved synthesis of PDOA (**2**) in only eight reaction steps, starting from known compounds, with 13% overall yield. The key oxidative intramolecular cyclization reaction at the *ortho*-position of the phenol derivative can be a useful tool to prepare other alkaloid frameworks or polycyclic compounds. The study of the scope and limitations of this reaction, and its application in the synthesis of other alkaloids, is underway.

## 3. Materials and Methods

### 3.1. General

The following instruments were used to obtain physical data: a JEOL (Tokyo, Japan) JNM-ECZ500R/S1 (^1^H-NMR: 500 MHz, ^13^C-NMR: 125 MHz) spectrometer for ^1^H and ^13^C NMR data, using tetramethylsilane as an internal standard; a Waters (Milford, MA, USA) Xevo G2-XS Q-Tof mass spectrometer for ESI-TOF MS. Microwave reactions were performed using Biotage (Uppsala, Sweden) Initiator+. Silica gel 60 N (Kanto (Tokyo, Japan) 63–210 μm) and pre-coated thin layer chromatography (TLC) plates (Merck (Darmstadt, Germany) 60F_254_) were used for column chromatography and TLC, respectively. Spots on the TLC plates were detected by spraying with an acidic *p*-anisaldehyde solution (*p*-anisaldehyde: 25 mL, *c*-H_2_SO_4_: 25 mL, AcOH: 5 mL, EtOH: 425 mL) or with a phosphomolybdic acid solution (phosphomolybdic acid: 25 g, EtOH: 500 mL) with subsequent heating. 7-(Benzyloxy)-6-methoxyisoquinoline (**5**) was prepared according to the literature [13]. Other chemicals were purchased from Sigma-Aldrich (St. Louis, MO, USA), Nacalai Tesque, Inc. (Kyoto, Japan), Tokyo Chemical Industry Co., Ltd. (Tokyo, Japan), or FUJIFILM Wako Pure Chemical Co. (Osaka, Japan). Hard copies of NMR spectra can be found as Appendix A.

### 3.2. Synthesis

#### 3.2.1. 7-(Benzyloxy)-6-methoxyisoquinoline 2-oxide (**6**)

7-(Benzyloxy)-6-methoxyisoquinoline (**5**, 2.11 g, 7.95 mmol) was dissolved in CH_2_Cl_2_ (80 mL). *m*CPBA (contains ca. 30% water) (2.23 g, 9.94 mmol, 1.25 equiv.) was added into the solution, and the whole mixture was allowed to be stirred for 30 min at rt. The reaction was quenched with saturated NaHCO_3_ aq. and extracted with CH_2_Cl_2_ for three times. The combined organic layer was dried over Na_2_SO_4_ and was concentrated in vacuo. The crude material was purified by silica gel column chromatography (EtOAc/MeOH = 4:1) to provide the product **6** (2.06 g, 92 %) as tan solid.

^1^H NMR (500 MHz, CDCl_3_) δ 8.57 (d, 1H, *J* = 1.9 Hz), 8.03 (dd, 1H, *J* = 7.0, 1.9 Hz), 7.50–7.46 (m, 3H), 7.41–7.38 (m, 2H), 7.36–7.32 (m, 1H), 7.06 (s, 1H), 7.01 (s, 1H), 5.25 (s, 2H), 4.00 (s, 3H). ^13^C NMR (125 MHz, CDCl_3_) δ 152.5, 151.3, 135.7, 135.1, 135.0, 128.9, 128.5, 127.5, 125.7, 125.6, 122.7, 105.5, 105.0, 71.1, 56.3. MS (ESI-TOF) *m*/*z*: 282 [M+H]^+^. HRMS (ESI-TOF) *m*/*z*: 282.1125, calcd for C_17_H_16_NO_3_^+^; Found: 282.1120.

#### 3.2.2. 7-(Benzyloxy)-6-methoxyisoquinoline-1(2H)-one (**7**)

**6** (2.82 g, 10.0 mmol) was dissolved in Ac_2_O (33.3 mL). Then the mixture was allowed to be heated at reflux with stirring for 3 h. The mixture was concentrated, and the residue was dissolved in 2 M NaOH aq. (66.7 mL) and was heated at reflux for 1 h. Then the mixture was acidified to pH 6 with 2 M HCl aq. and was extracted with CH_2_Cl_2_ for three times. The combined organic layer was dried over Na_2_SO_4_ and was concentrated in vacuo. The crude material was purified by silica gel column chromatography (CH_2_Cl_2_/EtOAc = 1:1) to provide the product **7** (1.60 g, 57 %) as pale brown solid.

^1^H NMR (500 MHz, CDCl_3_) δ 11.06 (br, 1H), 7.86 (s, 1H), 7.50–7.49 (m, 2H), 7.40–7.37 (m, 2H), 7.33–7.30 (m, 1H), 7.09 (dd, 1H, *J* = 7.1, 4.2 Hz), 6.92 (s, 1H), 6.48 (d, 1H, *J* = 7.0 Hz), 5.26 (s, 2H), 3.98 (s, 3H). ^13^C NMR (125 MHz, CDCl_3_) δ 163.6, 154.3, 148.5, 136.5, 134.1, 128.7, 128.2, 127.8, 126.6, 120.0, 108.7, 106.5, 106.3, 70.9, 56.2. MS (ESI-TOF) *m*/*z*: 282 [M+H]^+^. HRMS (ESI-TOF) *m*/*z*: 282.1125, calcd for C_17_H_16_NO_3_^+^; Found: 282.1120.

#### 3.2.3. 7-(Benzyloxy)-6-methoxyisoquinolin-1-yl Trifluoromethanesulfonate (**8**)

**7** (500 mg, 1.78 mmol) was dissolved in anhydrous CH_2_Cl_2_ (50 mL) under N_2_ atmosphere. Et_3_N (0.50 mL, 3.55 mmol, 2.0 equiv.) was added to the mixture and the whole mixture was cooled to –78 °C. Tf_2_O (0.36 mL, 2.22 mmol, 1.25 equiv.) was slowly added to the mixture and was stirred for 30 min at –78 °C. The reaction was quenched with saturated NaHCO_3_ aq. and was extracted with CH_2_Cl_2_ for three times. The combined organic layer was dried over Na_2_SO_4_ and was concentrated in vacuo. The crude material was purified by silica gel column chromatography (toluene/CHCl_3_ = 10:1) to provide the product **8** (671 mg, 91 %) as white solid.

^1^H NMR (500 MHz, CDCl_3_) δ 8.05–8.04 (d, 1H, *J* = 5.6 Hz), 7.53–7.49 (m, 3H), 7.42–7.38 (m, 2H), 7.36–7.32 (m, 2H), 7.14 (s, 1H), 5.30 (s, 2H), 4.04 (s, 3H). ^13^C NMR (125 MHz, CDCl_3_) δ 154.7, 151.9, 150.8, 138.5, 136.7, 135.6, 128.9, 128.5, 127.7, 120.8, 118.7 (q, *C*F_3_), 115.5, 106.2, 102.5, 71.2, 56.4. MS (ESI-TOF) *m*/*z*: 414 [M+H]^+^. HRMS (ESI-TOF) *m*/*z*: 414.0618, calcd for C_18_H_15_F_3_NO_5_S^+^; Found: 414.0604.

#### 3.2.4. 7-(Benzyloxy)-6-methoxy-1-vinylisoquinoline (**4**)

**8** (400 mg, 0.968 mmol), potassium vinyltrifluoroborate **5** (194 mg, 1.45 mmol, 1.5 equiv.), Pd(dppf)Cl_2_·CH_2_Cl_2_ (79.0 mg, 0.0968 mmol, 10 mol%), and Et_3_N (0.27 mL, 1.94 mmol, 2.0 equiv.) were dissolved in 2-propanol/THF (3:1) in a microwave vial and the vial was capped with an aluminum cap. Then the mixture was stirred at 80 °C under microwave irradiation for 6 h, concentrated. The reaction mixture was treated with H_2_O and was extracted with CH_2_Cl_2_ for three times. The combined organic layer was dried over Na_2_SO_4_ and was concentrated in vacuo. The crude material was purified by silica gel column chromatography (EtOAc/CH_2_Cl_2_ = 1:10) to provide the product **4** (221 mg, 78 %) as tan solid.

^1^H NMR (500 MHz, CDCl_3_) δ 8.38 (d, 1H, *J* = 5.6 Hz), 7.52–7.50 (m, 3H), 7.43–7.42 (m, 2H), 7.40–7.36 (m, 2H), 7.35–7.33 (m, 1H), 7.07 (s, 1H), 6.38 (dd, 1H, *J* = 16.9, 2.0 Hz), 5.64 (dd, 1H, *J* = 10.8, 2.0 Hz), 5.29 (s, 2H), 4.02 (s, 3H). ^13^C NMR (125 MHz, CDCl_3_) δ 153.1, 152.7, 149.2, 141.6, 136.3, 133.8, 132.6, 128.9, 128.3, 127.5, 122.3, 121.1, 119.2, 105.4, 105.3, 71.0, 56.1. MS (ESI-TOF) *m*/*z*: 292 [M+H]^+^. HRMS (ESI-TOF) *m*/*z*: 292.1332, calcd for C_19_H_18_NO_2_^+^; Found: 292.1318.

#### 3.2.5. 1-(2-Azidoethyl)-7-(benzyloxy)-6-methoxyisoquinoline (**9**)

**4** (231 mg, 0.793 mmol,), Cu(OTf)_2_ (14.3 mg, 0.0397 mmol, 5 mol%), tetra-*n*-butylammonium iodide (TBAI, 29.3 mg, 0.0793 mmol, 10 mol%), and *t*-butyl peroxybenzoate (TBPB, 0.23 mL, 1.189 mmol, 1.5 equiv.) were dissolved in H_2_O (3.2 mL) under N_2_ atmosphere. The mixture was heated at 50 °C with stirring. TMSN_3_ (0.16 mL, 1.189 mmol, 1.5 equiv.) was slowly added to the mixture and the whole mixture was stirred for 4 h at 50 °C. The reaction was quenched with saturated NaHCO_3_ aq. and was extracted with EtOAc for three times. The combined organic layer was dried over Na_2_SO_4_ and was concentrated in vacuo. The crude material was purified by silica gel column chromatography (EtOAc/hexane = 1:1) to provide the product **9** (225 mg, 85 %) as tan solid.

^1^H NMR (500 MHz, CDCl_3_) δ 8.31 (d, 1H, *J* = 5.7 Hz), 7.52–7.50 (m, 2H), 7.42–7.39 (m, 3H), 7.35–7.33 (m, 1H), 7.32 (s, 1H), 7.08 (s, 1H), 5.32 (s, 2H), 4.03 (s, 3H), 3.75 (t, 2H, *J* = 7.3 Hz), 3.38 (t, 2H, *J* = 7.3 Hz). ^13^C NMR (125 MHz, CDCl_3_) δ 155.1, 153.3, 149.2, 140.8, 136.3, 133.3, 128.9, 128.4, 127.4, 123.0, 118.9, 105.7, 105.6, 71.1, 56.2, 50.2, 34.1. MS (ESI-TOF) *m*/*z*: 335 [M+H]^+^. HRMS (ESI-TOF) *m*/*z*: 335.1503, calcd for C_19_H_19_N_4_O_2_^+^; Found: 335.1487.

#### 3.2.6. 1-(2-Azidoethyl)-6-methoxyisoquinoline-7-ol (**3**)

**9** (138 mg, 0.413 mmol) was dissolved in anhydrous CH_2_Cl_2_ (2.1 mL) under N_2_ atmosphere and the whole mixture was cooled to –78 °C. BCl_3_ (1 M in heptane, 1.03 mL, 1.03 mmol, 2.5 equiv.) was slowly added to the mixture and the whole mixture was stirred for 30 min at –78 °C. The reaction was quenched with saturated NaHCO_3_ aq. and was extracted with CH_2_Cl_2_ for three times. The combined organic layer was dried over Na_2_SO_4_ and was concentrated in vacuo. The crude material was purified by silica gel column chromatography (EtOAc/hexane = 10:1) to provide the product **3** (86.5 mg, 86%) as tan solid.

^1^H NMR (500 MHz, CDCl_3_) δ 8.30 (d, 1H, *J* = 5.7 Hz), 7.50 (s, 1H), 7.42 (d, 1H, *J* = 5.7 Hz), 7.01 (s, 1H), 4.03 (s, 3H), 3.83 (t, 2H, *J* = 7.3 Hz), 3.43 (t, 2H, *J* = 7.3 Hz). ^13^C NMR (125 MHz, CDCl_3_) δ 155.4, 151.0, 147.1, 140.2, 132.8, 123.7, 119.1, 106.8, 105.1, 56.2, 50.2, 34.1. MS (ESI-TOF) *m*/*z*: 245 [M+H]^+^. HRMS (ESI-TOF) *m*/*z*: 245.1033, calcd for C_12_H_13_N_4_O_2_^+^; Found: 245.1036.

#### 3.2.7. 6,7-Dimethoxyisoquinoline 2-oxide (**11**)

6,7-Dimethoxyisoquinoline (**10**, 95 mg, 0.50 mmol) was dissolved in CH_2_Cl_2_ (2.5 mL). *m*CPBA (contains ca. 30% water) (130 mg, 0.75 mmol, 1. 5 equiv.) was added into the solution, and the whole mixture was allowed to be stirred for 4 h at rt. The reaction was quenched with saturated NaHCO_3_ aq. and extracted with CH_2_Cl_2_ for three times. The combined organic layer was dried over Na_2_SO_4_ and was concentrated in vacuo. The crude material was purified by silica gel column chromatography (EtOAc/CH_2_Cl_2_/MeOH = 3:3:2) to provide the product **11** (91 mg, 89 %) as colorless oil.

^1^H NMR (500 MHz, CDCl_3_) δ 8.56 (s, 1H), 7.97 (dd, 1H, *J* = 7.0, 1.8 Hz), 7.43 (d, 1H, *J* = 7.0 Hz), 6.97 (s, 1H), 6.88 (s, 1H), 3.92 (s, 6H). ^13^C NMR (125 MHz, CDCl_3_) δ 152.1, 152.0, 134.9, 134.7, 125.6 (2C), 122.6, 105.1, 103.2, 56.2 (2C). MS (ESI-TOF) *m*/*z*: 206 [M+H]^+^. HRMS (ESI-TOF) *m*/*z*: 206.0812, calcd for C_11_H_12_NO_3_^+^; Found: 206.0818.

#### 3.2.8. 6,7-Dimethoxyisoquinolin-1-yl Trifluoromethanesulfonate (**13**)

**11** (1.28 g, 6.23 mmol) was dissolved in Ac_2_O (20.7 mL). Then the mixture was allowed to be heated at reflux with stirring for 1 h. The mixture was concentrated, and the residue was dissolved in 2 M NaOH aq. (41.6 mL) and was heated at reflux for 3 h. Then the mixture was acidified to pH 6 with 2 M HCl aq. and was extracted with CH_2_Cl_2_ for three times. The combined organic layer was dried over Na_2_SO_4_ and was concentrated in vacuo to give crude material containing compound **12**.

The crude product of the above reaction was dissolved in anhydrous CH_2_Cl_2_ (90 mL) under N_2_ atmosphere. Et_3_N (1.63 mL, 11.6 mmol) was added to the mixture and the whole mixture was cooled to –78 °C. Tf_2_O (1.20 mL, 7.40 mmol) was slowly added to the mixture and was stirred for 1 h at –78 °C. The reaction was quenched with saturated NaHCO_3_ aq. and was extracted with CH_2_Cl_2_ for three times. The combined organic layer was dried over Na_2_SO_4_ and was concentrated in vacuo. The crude material was purified by silica gel column chromatography (hexane/EtOAc = 3:1) to provide the product **13** (587 mg, 28 % in 2 steps) as white solid.

^1^H NMR (500 MHz, CDCl_3_) δ 7.99 (d, 1H, *J* = 5.6 Hz), 7.47 (d, 1H, *J* = 5.6 Hz), 7.20 (s, 1H), 7.08 (s, 1H), 4.00 (s, 6H). ^13^C NMR (125 MHz, CDCl_3_) δ 154.2, 151.8, 151.7, 138.4, 136.6, 118.7 (q, *C*F_3_), 120.8, 115.5, 105.0, 100.4, 56.3, 56.2. MS (ESI-TOF) *m*/*z*: 338 [M+H]^+^. HRMS (ESI-TOF) *m*/*z*: 338.0305, calcd for C_12_H_10_NO_5_FS^+^; Found: 338.0311.

#### 3.2.9. 6,7-Dimethoxy-1-vinylisoquinoline (**14**)

**13** (362 mg, 1.07 mmol), potassium vinyltrifluoroborate **5** (214 mg, 1.60 mmol, 1.5 equiv.), Pd(dppf)Cl_2_·CH_2_Cl_2_ (87.4 mg, 0.107 mmol, 10 mol%), and Et_3_N (0.30 mL, 2.14 mmol, 2.0 equiv.) were dissolved in 2-propanol/THF (3:1) in a microwave vial and the vial was capped with an aluminum cap. Then the mixture was stirred at 80 °C under microwave irradiation for 6 h, concentrated. The reaction mixture was treated with H_2_O and was extracted with CH_2_Cl_2_ for three times. The combined organic layer was dried over Na_2_SO_4_ and was concentrated in vacuo. The crude material was purified by silica gel column chromatography (hexane/EtOAc = 1:1) to provide the product **14** (178 mg, 77 %) as tan solid.

^1^H NMR (500 MHz, CDCl_3_) δ 8.37 (d, 1H, *J* = 5.5 Hz), 7.48 (dd, 1H, *J* = 16.9, 10.9 Hz), 7.40 (d, 1H, *J* = 5.5 Hz), 7.39 (s, 1H), 7.01 (s, 1H), 6.46 (d, 1H, *J* = 16.9 Hz), 5.66 (d, 1H, *J* = 10.8 Hz), 4.01 (s, 3H), 4.00 (s, 3H). ^13^C NMR (125 MHz, CDCl_3_) δ 152.6, 152.5, 150.1, 141.5, 133.6, 132.6, 122.4, 121.1, 119.3, 105.1, 102.9, 56.1, 56.0. MS (ESI-TOF) *m*/*z*: 216 [M+H]^+^. HRMS (ESI-TOF) *m*/*z*: 216.1019, calcd for C_13_H_14_NO_2_^+^; Found: 216.1025.

#### 3.2.10. 1-(Azidomethyl)-6,7-dimethoxyisoquinoline (**15**)

**14** (179 mg, 0.83 mmol,), Cu(OTf)_2_ (15.2 mg, 0.042 mmol), TBAI (30.7 mg, 0.083 mmol,), and TBPB (0.23 mL, 1.2 mmol) were dissolved in H_2_O (3.3 mL) under N_2_ atmosphere. The mixture was heated at 50 °C with stirring. TMSN_3_ (0.16 mL, 1.2 mmol) was slowly added to the mixture and the whole mixture was stirred for 3 h at 50 °C. The reaction was quenched with saturated NaHCO_3_ aq. and was extracted with EtOAc for three times. The combined organic layer was dried over Na_2_SO_4_ and was concentrated in vacuo. The crude material was purified by silica gel column chromatography (EtOAc/toluene = 1:1) to provide the product **15** (89.2 mg, 42 %) as pale-yellow solid.

^1^H NMR (500 MHz, CDCl_3_) δ 8.33 (d, 1H, *J* = 5.6 Hz), 7.42 (d, 1H, *J* = 5.6 Hz), 7.29 (s, 1H), 7.07 (s, 1H), 4.04 (s, 3H), 4.03 (s, 3H), 3.91 (t, 2H, *J* = 7.2 Hz), 3.48 (t, 2H, *J* = 7.2 Hz). ^13^C NMR (125 MHz, CDCl_3_) δ 155.1, 152.8, 150.3, 141.0, 133.2, 123.2, 118.9, 105.5, 102.9, 56.2 (2C), 50.2, 34.0. MS (ESI-TOF) *m*/*z*: 259 [M+H]^+^. HRMS (ESI-TOF) *m*/*z*: 259.1190, calcd for C_13_H_15_N_4_O_2_^+^; Found: 259.1200.

#### 3.2.11. *tert*-Butyl (2-(6,7-Dimethoxyisoquinolin-1-yl)ethyl)carbamate (**17**)

**13** (152 mg, 0.46 mmol), potassium β-(*N*-Boc)aminoethyltrifluoroborate **16** (172 mg, 0.67 mmol, 1.5 equiv.), Pd(dppf)Cl_2_·CH_2_Cl_2_ (37.6 mg, 0.046 mmol, 10 mol%), and K_2_CO_3_ (189 mg, 1.37 mmol, 3.0 equiv.) were dissolved in 1,4-dioxane (2.5 mL). Then the mixture was stirred at 80 °C for 6 h, and was concentrated. The reaction mixture was treated with H_2_O and was extracted with CH_2_Cl_2_ for three times. The combined organic layer was dried over Na_2_SO_4_ and was concentrated in vacuo. The crude material was purified by silica gel column chromatography (hexane/EtOAc = 1:1) to provide the product **17** (86 mg, 56 %) as tan solid.

^1^H NMR (500 MHz, CDCl_3_) δ 8.28 (d, 1H, *J* = 5.5 Hz), 7.46 (s, 1H), 7.39 (1H, d, *J* = 5.5 Hz), 7.04 (s, 1H), 5.42 (t, 1H, *J* = 6.1 Hz), 4.05 (s, 3H), 4.04 (s, 3H), 3.69 (t, 2H, *J* = 6.3 Hz), 3.39 (t, 2H, *J* = 6.3 Hz), 1.41 (s, 9H). ^13^C NMR (125 MHz, CDCl_3_) δ 156.6, 156.3, 152.8, 150.2, 140.6, 133.0, 122.3, 118.6, 105.3, 103.6, 79.2, 56.3, 56.2, 39.0, 35.0, 28.5 (3C). MS (ESI-TOF) *m*/*z*: 333 [M+H]^+^. HRMS (ESI-TOF) *m*/*z*: 333.1809, calcd for C_18_H_25_N_2_O_4_^+^; Found: 333.1821.

#### 3.2.12. Reaction of Compound **17** to **15**

**17** (30 mg, 0.091 mmol) was dissolved in CH₂Cl₂ (1.08 mL). TFA (0.14 mL, 20 equiv.) was added to the solution at 0 °C and the whole mixture was stirred for 5 h at rt to give crude **18**, which was dissolved in MeOH (0.45 mL). **19** (12.5 mg, 1.1 mmol, 12 equiv.), K_2_CO_3_ (16.7 mg, 1.8 mmol, 2.0 equiv.), and CuSO_4_·5H_2_O (0.15 mg, 0.9 μmol, 0.01 equiv.) were successively added to the solution and the whole mixture was stirred for 3 h at rt. The mixture was concentrated in vacuo to give crude product, which was purified by silica gel column chromatography (CHCl₃/MeOH/H₂O = 150:3:1) to provide the product **15** (12.3 mg, 74%). 

#### 3.2.13. Demethyl(oxy)aaptamine (**1**)

**3** (30.0 mg, 0.123 mmol) was dissolved in 1,1,1,3,3,3-hexafluoro-2-propanol (HFIP, 2.46 mL) and the whole mixture was cooled to 0 °C. Iodobenzene diacetate (PIDA, 68.9 mg, 0.246 mmol, 2.0 equiv.) was slowly added to the mixture and the whole mixture was stirred for 12 h at rt. The reaction was quenched with saturated NaHCO_3_ aq. and saturated Na_2_S_2_O_3_ aq., and the whole mixture was extracted with CH_2_Cl_2_ for three times. The combined organic layer was dried over Na_2_SO_4_ and was concentrated in vacuo. The crude material was purified by silica gel column chromatography (CH_2_Cl_2_/MeOH = 30:1) to provide the product **1** (18.3 mg, 70 %) as green solid.

^1^H NMR (500 MHz, CDCl_3_) δ 9.17 (d, 1H, *J* = 5.5 Hz), 9.09 (d, 1H, *J* = 4.4 Hz), 8.17 (d, 1H, *J* = 5.6 Hz), 7.48 (d, 1H, *J* = 4.4 Hz), 6.70 (s, 1H), 4.00 (s, 3H). ^13^C NMR (125 MHz, CDCl_3_) δ 178.0, 156.9, 156.4, 149.6, 149.5, 147.6, 136.8, 127.1, 121.7, 118.3, 108.2, 56.4. MS (ESI-TOF) *m*/*z*: 213 [M+H]^+^. HRMS (ESI-TOF) *m*/*z*: 213.0659, calcd for C_12_H_9_N_2_O_2_^+^; Found: 213.0658.

#### 3.2.14. 3-(Phenethylamino)demethyl(oxy)aaptamine (**2**)

**1** (10.3 mg, 0.049 mmol) was dissolved in H_2_O (0.12 mL). 2-Phenethylamine (0.018 mL, 0.147 mmol, 3.0 equiv.) was added to the mixture and the whole mixture was stirred for 24 h under air. Saturated NaHCO_3_ aq. was added to the mixture and the whole mixture was extracted with CH_2_Cl_2_ for three times. The combined organic layer was dried over Na_2_SO_4_ and was concentrated in vacuo. The crude material was purified by silica gel column chromatography (MeOH/CH_2_Cl_2_ = 1:25) to provide the product **2** (12.3 mg, 76%) as red solid.

^1^H NMR (500 MHz, CDCl_3_) δ 8.74 (d, 1H, *J* = 4.5 Hz), 8.39 (s, 1H), 7.45 (d, 1H, *J* = 4.5 Hz), 7.36–7.32 (m, 2H), 7.28–7.26 (m, 3H), 7.00 (t, 1H, *J* = 6.3 Hz), 6.60 (s, 1H), 3.96 (s, 3H), 3.79 (q, 2H, *J* = 7.3 Hz), 3.11 (t, 2H, *J* = 7.3 Hz). ^13^C NMR (125 MHz, CDCl_3_) δ 176.1, 157.9, 150.9, 144.0, 138.0, 136.4, 136.3, 134.7, 129.7, 129.0, 128.9, 127.1, 121.8, 118.0, 106.4, 56.2, 44.3, 35.5. MS (ESI-TOF) *m*/*z*: 332 [M+H]^+^. HRMS (ESI-TOF) *m*/*z*: 332.1394, calcd for C_20_H_18_N_3_O_2_^+^; Found: 332.1379.

## Data Availability

All data generated or analyzed during this study are included in this article or Appendix A.

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
