# Peer review of "Oxidative Cyclization at *ortho*-Position of Phenol: Improved Total Synthesis of 3-(Phenethylamino)demethyl(oxy)aaptamine"

_marinedrugs, 2023, doi:10.3390/md21050311_

Round 1

Reviewer 1 Report

The manuscript “Oxidative Cyclization at ortho-Position of Phenol: Improved Total Synthesis of 3-(Phenethylamino)demethyl(oxy)aaptamine” describes the synthesis of the demethyl(oxy)aaptamine skeleton via oxidative intramolecular cyclization of 1-(2-azidoethyl)-6-methoxyisoquinolin-7-ol followed by dehydrogenation with a hypervalent iodine reagent. The manuscript was professionally produced and should be published in the present state. Suggestions for the authors to consider are listed below.

 Scheme 3: Synthesis of demethyl(oxy)aaptamine (1) and PDOA (2).

Suggestion: Please include structure of compound 2 in Scheme 3.

Note: Reader needs to go to page 1 to find structure 2.

 Scheme 4: Plausible reaction mechanism of the oxidative cyclization of 3 to 1.

Suggestion: Please include structure of compound 1 in Scheme 4.

Note: Reader needs to go to page 1 to find structure 1.

 Scheme 4: Plausible reaction mechanism of the oxidative cyclization of 3 to 1.

Suggestion: Plausible reaction mechanism for the oxidative cyclization of 3 to 1.

The manuscript was professionally produced and should be published in the present state.

Author Response

Thank you for your generous consideration. Our revisions and comments to the suggestions from the reviewer were listed below.

Point 1. Scheme 3: Synthesis of demethyl(oxy)aaptamine (1) and PDOA (2).; Please include structure of compound 2 in Scheme 3.Reader needs to go to page 1 to find structure 2.

Response 1. Structure of compound 2 was included.

Point 2. Scheme 4: Plausible reaction mechanism of the oxidative cyclization of 3 to 1.; Please include structure of compound 1 in Scheme 4. Reader needs to go to page 1 to find structure 1.

Response 2. Structure of compound 1 was included.

Point 3. Scheme 4: Plausible reaction mechanism of the oxidative cyclization of 3 to 1. -> Plausible reaction mechanism for the oxidative cyclization of 3 to 1.

Response 3. Revised as recommended.

Reviewer 2 Report

Review Report

Oxidative Cyclization at ortho-Position of Phenol: Improved 2 Total Synthesis of 3-(Phenethylamino)demethyl(oxy)aaptamine

A shorter synthesis of the demethyl(oxy)aaptamine skeleton was developed via oxidative intramolecular cyclization of 1-(2-azidoethyl)-6-methoxyisoquinolin-7-ol followed by dehydrogenation with a hypervalent iodine reagent. This is the first example of oxidative cyclization at the orthoposition of phenol that does not involve spiro-cyclization, resulting in the improved total synthesis of 3-(phenethylamino)demethyl(oxy)aaptamine, a potent anti-dormant mycobacterial agent.

1.       The authors have well established the total synthesis of 3-(Phenethylamino)demethyl(oxy)aaptamine however, they should expand the introduction part with current references.

2.       In line no 57, there must be a coma before in this study…..

3.       In line no 76, Remove word and before 8

4.       Authors should explain the 1H and 13C NMR interpretations of reported intermediates in the supporting information. Moreover, the -OMe group having 3 protons but authors shown 3.9 (4) protons, as my experience, its not analytically acceptable. Hope, they should correct them carefully.

5.       In Table 1, did authors study 0.1 conc. of HFIP for 5 h to enhance the yield %?

6.       Authors should cite suitable reference for the Scheme 4 mechanism from the literature?

7.       In summary, authors mentioned the overall yield is just 13%? Is it acceptable in synthetic chemistry? Justify?

Result : Major Revision.

Dr. Ravi Kumar Cheedarala,

Changwon National University,

S. Korea.

yes

Author Response

Thank you for your generous consideration. Our revisions and comments to the suggestions from the reviewer were listed below.

Point 1. The authors have well established the total synthesis of 3-(Phenethylamino)demethyl(oxy)aaptamine however, they should expand the introduction part with current references.

Response 1. Some references about synthesis and bioactivities of aaptamine and its derivatives were added to expand the introduction.

Point 2.      In line no 57, there must be a coma before in this study…..

Response 2. The words, "Based on the above background" were not needed but we forgot to delete them before submission. They were deleted in the revision.

Point 3.     In line no 76, Remove word and before 8

Response 3. It does not mean that potassium vinyltrifluoroborate is compound 8. In this reaction, potassium vinyltrifluoroborate and compound 8 were coupled to yield compound 4. So, the word "compound" was added before 8.

Point 4.     Authors should explain the 1H and 13C NMR interpretations of reported intermediates in the supporting information. Moreover, the -OMe group having 3 protons but authors shown 3.9 (4) protons, as my experience, its not analytically acceptable. Hope, they should correct them carefully.

Response 4. We think the reviewer might misunderstand the chemical shift value of the 1H NMR signal of -OMe group as integration value. Chemical shift values are shown above the signals, and integration values are shown below the signals.

Point 5.     In Table 1, did authors study 0.1 conc. of HFIP for 5 h to enhance the yield %?

Response 5. The reaction in the higher concentration such as 0.1 M resulted in the lower yield because of the production of various side products. They might be derived from intermolecular oxidative coupling, although their detailed structures have not been analyzed. Longer reaction period does not enhance the yield. The following sentences were added to section 2.2.2. before "The use of highly ...".

The reaction in higher concentration (0.1 M) provided various side products, probably through intermolecular coupling reaction, resulting in the lower yield, and the use of ...

Point 6.     Authors should cite suitable reference for the Scheme 4 mechanism from the literature?

Response 6. The suitable references were added.

Point 7.     In summary, authors mentioned the overall yield is just 13%? Is it acceptable in synthetic chemistry? Justify?

Response 7. As the overall yield is multiplication of the respective reactions, it is reasonable from the viewpoint of significant digit. Many papers about synthetic chemistry use similar expression.

Reviewer 3 Report

The manuscript of Kotoku et al entitled “Oxidative Cyclization at ortho-Position of Phenol: Improved Total Synthesis of 3-(Phenethylamino)demethyl(oxy)aaptamine” describes a new protocol to synthesize aaptamine. Although there are several methodologies described to synthesize this heterocycle, the strategy presented by the authors is very interesting and promising. I recommend the following minor revisions.

1. The authors have commented “There are numerous reports on the total synthesis of aaptamines [3]”. However, this reference, appropriately added, is from 2009. I was wondering if any other synthesis of this heterocycle has been published since then (for instance, see Org. Lett. 2019, 21, 1430−1433). Could the authors check this? If any protocol has been presented after 2009, please, include the references.

2. Figure 2 has been not cited in the text. Therefore, in line 42, I suggest the author cite Figure 2 at the end of “… quinoline (AC) rings.”

3. Could the authors include a reference for “. This suggests that the reaction … activation proposed here.” (lines 117-119)

4. Concerning Table 1, please inform whether it is an isolated yield, and the amount in mmol terms of each compound.

Author Response

Thank you for your generous consideration. Our revisions and comments to the suggestions from the reviewer were listed below.

Point 1. The authors have commented “There are numerous reports on the total synthesis of aaptamines [3]”. However, this reference, appropriately added, is from 2009. I was wondering if any other synthesis of this heterocycle has been published since then (for instance, see Org. Lett. 2019, 21, 1430−1433). Could the authors check this? If any protocol has been presented after 2009, please, include the references.

Response 1. Some recent papers about the total synthesis of aaptamine and its derivatives were added to the reference. The suggested paper, Org. Lett. 2019, 21, 1430−1433, has already been included in the reference, but the order was changed.

Point 2.      Figure 2 has been not cited in the text. Therefore, in line 42, I suggest the author cite Figure 2 at the end of “… quinoline (AC) rings.”

Response 2. Thank you for pointing it out. The citation was added.

Point 3.     Could the authors include a reference for “. This suggests that the reaction … activation proposed here.” (lines 117-119)

Response 3. The suitable references were added.

Point 4.     Concerning Table 1, please inform whether it is an isolated yield, and the amount in mmol terms of each compound.

Response 4. The yields are isolated yield, and the reactions were conducted in 0.1 mmol scale. These were added in the Table footnote.

Round 2

Reviewer 2 Report

Accept the present format of the manuscript.